# Perceptions of a State-Level HPV Vaccine Mandate and Exemption Option in Rural Virginia: A Qualitative Study

**DOI:** 10.3390/vaccines12040401

**Published:** 2024-04-10

**Authors:** E. Marshall Brooks, Kendall Fugate-Laus, Ben Webel, Shillpa Naavaal

**Affiliations:** 1Department of Family Medicine and Population Health, Virginia Commonwealth University, Richmond, VA 23284, USA; 2Dental Public Health and Policy, School of Dentistry, Virginia Commonwealth University, Richmond, VA 23298, USA

**Keywords:** HPV vaccine, vaccine coverage, vaccine acceptance, health policy, community perspective, parental attitudes, provider attitudes, school-entry vaccine requirements, vaccine mandates

## Abstract

Human papillomavirus (HPV) is the most common sexually transmitted infection in the United States; yet, despite the availability of safe and effective HPV vaccines, only half of eligible adolescents have completed the vaccine series. School-entry requirements are one proven strategy to increase vaccination rates among children and youth and reduce the burden of HPV-related cancer. This study investigated community perceptions of an HPV vaccine school-entry mandate in Virginia and the consequences of a low threshold exemption option included in the legislation. We conducted 40 interviews with community members including 15 interviews with parents, 19 with healthcare providers, and 6 with community leaders. Interviews asked about knowledge, beliefs, and attitudes concerning the HPV vaccine and mandate. Interviews were recorded, transcribed, and thematically analyzed. Despite healthcare provider support for the mandate, there was widespread confusion over the school-entry policy and concern that the exemption option undermined vaccination efforts. Understanding variations in community-level perceptions and response to school-based vaccination mandates is crucial for designing effective public health strategies. Findings suggest statewide vaccination initiatives should preemptively identify low uptake areas and provide targeted information to communities. Future mandates should avoid the use of ambiguous and contradictory language in vaccine-related legislation.

## 1. Introduction

Human papillomavirus (HPV) is the most common sexually transmitted infection in the United States, causing nearly 90% of cervical cancer cases, as well as a significant proportion of other cancers (including oropharyngeal cancer, anal cancer, vulvar cancer, vaginal cancer, and penile cancer), totaling nearly 45,000 cases a year [1]. The Advisory Committee on Immunization Practices (ACIP) has recommended that females (since 2006) and males (since 2011) between the ages of 11 and 12 receive the HPV vaccine as a routine vaccination which can be given as early as age 9 [2]. However, despite the availability of safe and effective HPV vaccines for over a decade, coverage in the U.S. has failed to meet the Healthy People 2020 goal of 80% series completion, with only 63% of adolescents aged 13–17 years having completed the vaccine series [3].

School-entry requirements have proven to increase vaccination rates among children and youth and reduce the burden of HPV-related cancer [4]. When vaccines are mandated, they become part of routine healthcare, helping normalize vaccines and make them less likely to be stigmatized or viewed as optional. Previous studies have shown that, when implemented effectively, school-based HPV vaccination programs in the United States were associated with a 63% decrease in cervical cancer incidence among girls and women aged 15–24 years [5]. Virginia is one of only a few states (along with Rhode Island, Puerto Rico, Hawaii, and the District of Columbia) that require HPV vaccination for school entry [6]. When first enacted in Virginia in 2009, this policy was required only for females, but in July 2021 a statewide mandate began requiring all incoming 7th graders (males and females) to receive the HPV vaccination. Despite these legislations, some studies have documented the ineffectiveness of HPV vaccine school-entry policies [7,8]. Furthermore, data show a wide variation in the HPV vaccination rates by geography, with significantly lower vaccination rates in rural areas [9,10].

Only one other qualitative study has evaluated parents’ perceptions of the HPV vaccination school-entry policy in Virginia and potential factors contributing to low vaccine uptake. Pitt et al. 2013 found that resistance to the HPV vaccine primarily stemmed from perceived threats to parental autonomy posed by the government mandated vaccine, followed by concerns about vaccine safety [11]. Similar concerns have been echoed in more recent studies, where the compounding influence of social and religious conservativism has also been noted [12]. As other quantitatively focused studies have noted, the Virginia HPV vaccine mandate has been less successful, in part due to an informed exemption option written into the mandate which differentiates the HPV vaccination mandate from other similar vaccines required for school entry [13,14,15,16]. Specifically, the Virginia Code Section 32.1-46 (D) (3) acknowledges that HPV is not easily transmitted within schools, and thus allows parents to exempt or “opt-out” their child from receiving the vaccine after reviewing State Board of Health-approved materials explaining the association between HPV and cervical cancer (Virginia Legislation Code, 2010, 2021) [17]. More research at the intersection of community experience of vaccine mandates and public health policy is, therefore, much needed to improve HPV vaccination.

Virginia offers a unique case to explore community perceptions of an HPV vaccination state mandate and an informed exemption option, as well as implications for other states considering similar actions. Previous studies on HPV vaccine mandates have either been policy-focused or exclusively drawn on quantitative data, leaving a qualitatively informed understanding of these issues largely unaddressed in the broader literature, and acutely so within the context of Virginia. Further, there has been a lack of focused inquiries on geographical areas of low uptake where future public health and policy efforts are needed. The purpose of this qualitative study is thus to identify and describe how community members perceive the HPV vaccine school mandate and the consequences of a vaccine opt-out.

## 2. Methods

This study is part of a larger mixed-method project designed to investigate HPV vaccine hesitancy in Virginia. For this paper, we specifically focus on the school mandate policy and participants’ perceptions of the mandate, concentrating on three rural counties in Virginia which rank among the highest in the state with respect to incidence and mortality rates for oral and cervical cancers and with respect to very low HPV vaccine initiation and completion rates for both males and females [10]. The three included counties were selected because they had very low HPV initiation (21–41%) and completion rates (10–20%); they fall under our cancer center catchment area, are rural, and are proximal to each other. All participants were recruited using a convenience and snowball sampling technique. Given the exploratory nature of our study, we sought to include 5–6 subjects for each of our primary subgroups, parent/guardians, healthcare provider, and community leaders, with in-depth experience living and working in the community. Parents and healthcare providers who participated in the survey arm of the project were asked if they would be interested in participating in the interviews. Those who showed interest in participating were contacted by phone or email and, if eligible, were scheduled for interviews. Parents were eligible if they resided in one of the counties of interest and had a child between 9 and 17 years of age. Healthcare providers were eligible if they were practicing in the state of Virginia and providing care to children and adolescents between the ages of 9 years and 17 years. The third group of participants consisted of community leaders, including public health officials, and school system officials who were identified from public sources and local partners. These individuals were contacted to determine interest in participating in the current study and were eligible if they served in one of the three counties included in the study. Interviews were conducted between March 2021 and February 2022.

A semi-structured interview guide was developed through a review of the relevant literature and discussion with subject matter experts. All participants were asked about knowledge, beliefs, and attitudes concerning HPV and the HPV vaccine, as well as their perceptions of barriers and facilitators to vaccine uptake in their region. Healthcare providers were asked additional questions focused on their experience engaging with patients with respect to HPV and the HPV vaccine in their clinical practice. All study authors were involved in conducting interviews, which were held primarily over Zoom to accommodate availability and schedules. All interviews were audio-recorded with the permission from the interviewee, de-identified, and transcribed verbatim using Rev, a web-based transcription service, and then spot-checked for accuracy. Interviews were then thematically coded using a mix of a priori concepts derived from the interview guide and relevant literature [18] as well as through an inductive immersion–crystallization approach to identify emergent themes [19]. First, two of the team members, i.e., a qualitative researcher (MB) and a PhD student (KFL), conducted thematic template-based coding using the a priori codebook. All transcripts were also reviewed by the principal investigator (SN) and feedback was provided throughout. After coding, both MB and KFL reviewed the transcripts first separately and then together to determine emergent codes and reach consensus for common definitions and appropriate use [20]. Each interview was evaluated separately during the coding process, and then combined to classify codes by theme. Differences were resolved through a consensus-building process conducted via team meetings. We determined data saturation by assessing when the data in transcripts began to repeat itself, and when we felt we had achieved a sufficient understanding of the identified themes (or codes) to address our primary research questions, thus rendering additional interviews redundant [21]. All study procedures and documents were approved by the Institutional Review Board of the Virginia Commonwealth University.

## 3. Results

We conducted a total of 40 interviews with community members, including 15 interviews with parents/guardians of HPV vaccine eligible children, 19 interviews with healthcare providers (including pediatricians, nurse practitioners, nurses, and dentists), and six interviews with community leaders (including public health officials, school board members, and school principals). The average interview length was 40 min. Findings are organized into two primary domains, each with multiple sub-themes, focused on (1) community members’ general perceptions of the HPV vaccine mandate and (2) the consequences of including an informed exemption option in the legislation (see Table 1). 

### 3.1. How Community Members Made Sense of the HPV Vaccine Mandate

#### 3.1.1. Support for Vaccine Mandates among Healthcare Providers

The healthcare providers we spoke with overwhelmingly supported the use of vaccine mandates as a necessary and effective means of promoting vaccine uptake and improving public health. When mandated by the state for school entry, conversations about the need for vaccination with parents and patients were reportedly made straightforward and routine. As one public health nurse stated, “Parents will say yes when it’s school required. Sometimes you don’t even have to explain what this is for”. We were told this is especially the case when multiple vaccines are combined and administered in a single shot. As another pediatrician reported, even conversations with vaccine-hesitant families are made uncomplicated under these conditions. “I’m glad to be in a state where it is mandated. I say, ‘Well, it’s one of those that he’s going to need to have before he can start seventh grade. We can talk about it again a little bit later, but this is one that will be necessary’”. Beyond streamlining conversations with patients and families, clinicians emphasized the public health significance of vaccine mandates, with one Virginia Department of Health (VDH) clinician remarking that, “The only way I know to make a real change is for schools to require vaccines, and when they require them, all of a sudden we have a herd immunity level”.

#### 3.1.2. Parents Resistant to Coercion but Open to Information

While mandates are often seen as a panacea for low vaccine uptake, a recurring concern heard across respondent groups was that school mandates could also inadvertently promote greater resistance to the vaccine, especially among parents who view it as “forcing their hand”. As one parent told us, “I can actually see people saying, ‘Now wait a minute. It wasn’t required last year or the year before last. Why are you now going to make my child take this? I don’t know if I really want to do it now because now you’re making me do it’”. This resistance, as one school nurse reported, is part of “the age we are in” in which people increasingly “don’t want to be told what to do”. A parent echoed this perspective, stating:

A lot of the people don’t want the government telling them what to do. They just don’t want somebody saying, “You have to have this before you go to work”. Or, “You have to have this before you can go to school”. And they don’t like the government… They want to make their own decisions; and they don’t want Uncle Sam telling them they have to do it. 

Concerns about autonomy and government coercion were contrasted with parents also expressing desire for additional information about the risks of infection and benefits of vaccination. It was suggested that informational materials should be provided, rather than making generic statements about “preventing cancer”, including statistical evidence that clearly demonstrates the relative risk of developing cancer. As one parent suggested, “If you could say to me as a parent, ‘Without this vaccine your child is X percent more likely to get cancer’, and put some actual real numbers, that would go a long way with convincing me”.

Such information, however, must also be forthright and transparent about the relative benefits and risks of vaccination. As one parent suggested, “I think there needs to be real openness so that we don’t try to paint it as a miracle drug”. Here, parents expressed concern about drug advertisements and education materials that appear to present a one-sided view of vaccination benefits. As another parent suggested, 

I would like to know more about, what are the risks? I don’t mean the risk of getting some form of cancer, I’m talking about the risk of getting the shot or the vaccine. If I get the vaccine, what might happen? I want to know, honestly, this might happen to you, so I can then determine, yes, it might happen, but the risk is worth it. 

#### 3.1.3. ”Requiring It as a Condition of Staying in School May Not Actually Serve a Purpose”

In addition to pushback on more overt political grounds, we also heard resistance to the HPV mandate from school officials who, while supporting the idea of children receiving the HPV vaccine, questioned the logic of requiring it as a condition for school entry.

“School is not all about health procedures and making sure that everyone gets vaccinated, that’s not our purpose. Our purpose is to provide education… Saying that you have to get the vaccine at school is kind of a catch-22. There’s no mechanism to compel this person to get vaccinated. They just would cease to be a student at school. So I think that I’m having a little conflict because the lens that I look at this through is the health, safety, and welfare of students while they’re on campus”.

From this perspective, the priority is to ensure the health of students while on campus, which means mitigating only those risks that pose an immediate danger to students while they occupy that particular environment. Here, the school official compared the relatively restricted modes of transmitting HPV to other infectious diseases that are more readily communicable in school settings.

I mentioned the swine flu vaccine earlier. I understood that to be something that children could routinely spread amongst each other at school. COVID, same way. It’s important for students to have that because they could get it. That Tdap shot. Yes. Children can spread whooping cough amongst each other at school. The HPV vaccine, I think, is a little different because we’re not talking about something that can readily be spread from one child to another just from casual contact within the same classroom.

#### 3.1.4. Confusion over Whether Recommended or Required

There was widespread confusion about whether the mandate made HPV vaccination a recommendation or a requirement. County health officials and school nurses we spoke with echoed this confusion, describing it ambiguous as “one of those optional mandated vaccines”.

So, HPV is, I forget how the law is written, but it’s not mandated that they get HPV [vaccine]. I think that’s how the literature is written or the state law is written. So right now it is not a mandated vaccine that students need for seventh grade; parents can opt out of getting HPV.

Indeed, the VDH website that lists the various vaccines needed for school entry evidences a vague and conflicting use of verbiage to describe the mandate. While HPV is listed as “required”, the legislation also stipulates that the child’s parent may unilaterally decide, even without seeking a medical or religious exemption, to indefinitely defer getting the vaccine. As stated on the VDH website, “After reviewing educational materials approved by the Board of Health, the parent or guardian, at the parent’s or guardian’s sole discretion, may elect for the child not to receive the HPV vaccine” [17]. This has predictably led to consternation among school nurses who are tasked with tracking HPV vaccination status among eligible students and nominally enforcing the mandate.

I typically say HPV is required along with measles, mumps, rubella, polio. And then if the client has heard of HPV before they’ll say, “Well, HPVs not… We don’t have to get that, right?” And some nurses will say, “Well actually it’s required”. But other nurses will be like, “Well, it’s required, but typically they won’t kick you out of school for not getting it, they’ll let you start school if you don’t have it”. Whereas that’s not the case with MMR or hepatitis B or the other ones. So they’re still allowed in school if they don’t have HPV. And in fact, it says on the [student] record, “Not required”. But the policy says “required”. So it’s a lot of inconsistencies there.

### 3.2. Consequences of a Vaccine “Opt-Out”—Healthcare Provider’s and Community Leader’s Perspectives

#### 3.2.1. Undermining HPV Vaccination Initiatives

Respondents described several consequences of the HPV vaccine opt-out and its impact on public health. Because the HPV vaccine is perceived as intrinsically “optional”, the mandate is reportedly minimally enforced. As one school employee stated, “when we send out the parent letter, it does have the HPV written on it. But the school does not push it as being ‘you have to have this in order to come to school’, because you don’t”. Similarly, when we talked to a county health official, they expressed frustration that opting out of the vaccine required no special action, and that the mandate was easily circumvented.

If you don’t have HPV [i.e., proof of vaccination on the student record], they don’t do anything about it. And then also, say you’re missing a vaccination, the clients will come in with a letter from the school that will list all their vaccines, how many they’ve had, what’s missing. And then there’s a final column that’ll say, required or not required. And it says “HPV not required”. I am frustrated about the school’s attitude toward HPV because they don’t really enforce it. I mean, no one’s going to be kicked out or sent away if they don’t get HPV.

According to one school official, this has consequently undermined vaccination efforts in their community. “I ask parents, is your child vaccinated for HPV? Everybody that I spoke to said no…The people have told me it’s not required so I’m not going to give my child the shot. That’s been the consensus of everybody, that it’s not required, therefore, we’re not going to do it”. A pediatrician we spoke with raised concerns about the health effects of this inconsistent and contradictory messaging, stating that some parents think that “it’s not mandatory, so it’s not important for the child to get it”.

Beyond just impacting individual parent/guardian-level decision making, a clinician reported that some local healthcare organizations also now no longer promote the HPV vaccine when providing other routine vaccinations to school age children in an effort to avoid negative interactions with parents.

Some of these acute care settings—Walgreens, different urgent care spots that people can go in really quick at the last minute and get a vaccination and come back to school and be fine with allowing entry into public school—I think a lot of those facilities are missing the mark on supporting that vaccination.

Because of this, the clinician suggested, efforts to educate parents on the risks of HPV and the benefits of vaccination were being effectively undermined as a part of standard care for those receiving vaccines outside of traditional pediatric/primary care settings. They expressed concern that this has not only resulted in missed opportunities to offer the vaccine to eligible people, but also detracted from public health messaging about the importance of vaccination, as parents may falsely conclude that if it were actually important, these organizations would actively promote it.

#### 3.2.2. Politics Dictate Public Health

Another consequence of the exemption option we heard about is that school officials who either personally disagree with the mandate or want to avoid controversy routinely rebuff local health department efforts to provide HPV-related education, including offers to host back-to-school vaccine drives. As a health department employee explained, “We’ve had a couple schools that won’t allow us to do that, because it’s an optionally mandated vaccine, so we’re not able to offer it in those school systems”. When asked why school districts are refusing these offers, one school board member reported that local elected officials’ with more “conservative” political ideologies often swayed public health decision making. They explained, “We are a pretty conservative district. I think it depends a lot on the school board members and their beliefs and their age range. I think that’s got a lot to do with it, and their own knowledge base. And politics comes into it at some point”.

#### 3.2.3. HPV as a Public Health Blind Spot

Partly attributable to the aforementioned issues, respondents described an overall lack of HPV-related information disseminated by or through the school system. “I have not heard of any push” one county health clinician reported, “no type of education, no pamphlets that are given out. I had one mother ask me what kind of vaccine?”. The same respondent saw this as part of a broader blind spot in the local public health system. “Nobody is really educating on HPV here. I mean, I’m pretty positive that’s not happening anywhere”. In addition to the lack of interest or concern about HPV vaccination, local school systems were also perceived as generally underperforming academically and thus primarily focused on maintaining their accreditation status. As one former school board member reported, “Three of our four schools have lost their accreditation. The top of their list is getting reaccredited, it’s not HPV prevention”.

## 4. Discussion

This study investigated community members’ perceptions of an HPV vaccination state mandate and the consequences of a low threshold vaccine exemption option included in the legislation. Specifically, we investigated how parents, healthcare providers, and community leaders understood and responded to the vaccine mandate and its opt-out allowance. While we found that healthcare providers supported the use of vaccine mandates as a necessary and effective means of promoting vaccine uptake, some parents and community leaders expressed concern that mandates may paradoxically increase the public’s resistance to getting the vaccine. There was similarly widespread confusion among healthcare providers and community leaders about the exact nature of the mandate, and school nurses were uncertain about their role in promoting or enforcing the vaccine requirement. Our study findings have significant implications for improving vaccine acceptance through the use of state mandates, with the potential to inform future HPV-related legislation. To enhance the uptake of vaccines and improve health outcomes, our study highlights key areas needing improvement with respect to how school mandates are written, disseminated, and enforced.

Support for vaccine mandates was high among healthcare providers, who underscored the ability of mandates to normalize vaccination as part of providing routine preventive care and streamline parent/guardian conversations, even with those predisposed to vaccine hesitancy. However, consistent with previous research, we heard significant concern that HPV vaccine mandates could inadvertently lead to greater resistance among parents who perceive these mandates as a form of government coercion [22]. For such parents, parental autonomy in children’s health care decisions overrides public health priorities or initiatives. Healthcare providers and school system leaders shared that this form of parental resistance was rising, especially amidst the COVID-19 pandemic, tracking broader societal trends among some populations to mistrust medical and public health professionals’ guidance on the receipt of recommended preventive services and vaccines [23].

School-based vaccination programs have the potential to significantly improve HPV vaccine coverage, but, as our findings underscore, they also face challenges and controversies. While Virginia became one of the first U.S. jurisdictions to implement HPV vaccine requirements in 2009, this legislation also uniquely allowed for broad philosophical exemptions [7]. As others have noted, this approach is believed to have inadvertently weakened the mandate’s effectiveness, as it facilitated parental exemptions and singled out the HPV vaccine from other vaccines with philosophical exemption policies [8,11,12,16]. Considering the mandate, we found widespread confusion among parents, school system leaders, and healthcare providers regarding whether the HPV vaccine is in fact mandatory (i.e., required for school entry) or simply recommended, and thus imminently optional. Here, there is a marked inconsistency between policy and practice, where the language of vaccine requirements conflicts with the legislatively dictated lack of enforcement options. Previous research has indicated that this may crucially undermine public health vaccination efforts, as ambiguity in the language of the mandate, largely around the ability to “opt-out”, has led to elevated uncertainty among school employees about what vaccines are required for school entry, along with parental confusion about the risk of HPV infection and decisions to ultimately forgo vaccination for their children [9,24].

The HPV vaccine mandate opt-out provision also risks undermining HPV vaccination public messaging efforts and, consequently, the ability of healthcare providers and public health officials to increase uptake [24]. One example of this is inconsistency across healthcare settings in the list of vaccines ostensibly required for school entry. As previous studies have suggested, such inconsistencies in clinician–patient communication may confuse patients/parents and exacerbate public mistrust of medical guidance, thus contributing to public misconceptions regarding the risk of HPV and benefits of vaccination [8,13,25]. Secondly, politicized conflicts among school officials over the appropriate role of public health campaigns in public education has contributed to a lack of support for vaccination initiatives and education on HPV within the school system [26]. Beyond the immediate impact of jeopardizing school aged children’s risk of contracting HPV, this may further entrench widespread apathy and ignorance about the risks of HPV and benefits of vaccination and thwart efforts to reduce negative health outcomes from HPV over the long term [11,24].

The situation in Virginia underscores the critical role of government policies and comprehensive health education in shaping the rate of vaccine uptake [27]. Furthermore, it highlights the need, at a minimum, for more effective communication strategies to educate the public on the relative risks and benefits of HPV vaccination [28,29]. Understanding variations in community-level perceptions and response to school-based vaccination mandates is crucial for designing effective public health strategies that can reduce the incidence of HPV-related diseases and associated health disparities [30]. To reduce confusion, future mandates should more clearly and consistently specify the exact nature of the vaccine requirement and avoid the use of ambiguous and contradictory language in legislative documents and informational materials created for public dissemination. Additionally, statewide vaccination campaigns should preemptively identify low uptake areas and provide targeted, consistent, and coordinated information to local community leaders, health providers, and school-based organizations to promote acceptance.

To our knowledge, this is the first qualitative study to use a diverse, multilevel sample of community members to elicit perspectives on a state-level HPV vaccine mandate and the consequences of including an exemption option in the legislation. It is also the first to focus exclusively on rural communities with very low HPV initiation and completion rates. Nevertheless, our study has several limitations worth identifying. First, because of the relatively small sample size and the use of convenience sampling, the ability to generalize findings is limited. Similarly, because of the use of convenience sampling, there is the potential for bias in our sample, as those who volunteered to be interviewed may have had atypically strong views about the HPV vaccine. In addition, we did not analyze changes in vaccination rates following the 2021 mandate, either at the population-level or at the individual-level among the children of parents we spoke to, so we cannot speak to the relative impact of the mandate.

## 5. Conclusions

This study sought to understand the implications of an HPV vaccination state mandate and the perceptions about including a broad vaccine exemption in the legislation. Specifically, we examined how parents, healthcare providers, and school system leaders made sense of and responded to the mandate and the ability to opt-out of the vaccine requirement. School-entry vaccine requirements are a strategy to increase vaccination uptake and reduce gaps between vaccine groups, including the HPV vaccine. However, to enhance uptake of vaccines and improve positive health outcomes from similar mandates, policy makers must consider the impact of legislation that includes broad exemption options. These findings have significant implications for vaccine acceptance in the context of state mandates, with the potential to inform future HPV-related legislation.

## Figures and Tables

**Table 1 vaccines-12-00401-t001:** Community member perceptions of the HPV vaccine mandate and the consequences of including an informed exemption option.

Domains	Key Themes
**Making Sense of the HPV Vaccine Mandate**	
Support for vaccine mandates among healthcare providers	Healthcare providers supported the use of vaccine mandates as a necessary and effective means of promoting vaccine uptake and public health.
Parents resistant to coercion but open to information	Respondents expressed concern that mandates would increase resistance to the vaccine, especially among parents who view it as the government “forcing their hand”.
“Requiring it as a condition of staying in school may not actually serve a purpose”.	While supportive of HPV vaccination, some school officials questioned the logic of requiring it as a condition for school entry. Primary responsibility was to mitigate only those risks that pose an immediate danger to students while in school settings.
Confusion over whether recommended or required	Widespread confusion among healthcare providers and community leaders about whether the HPV vaccine was a recommendation, suggestion, or a requirement. School nurses were particularly uncertain about how best to promote or enforce the vaccine requirement.
**Consequences of a Vaccine “Opt-out”—Healthcare Providers’ and Community Leaders’ Perspectives**	
Undermining HPV vaccination initiatives	Healthcare providers reported that confusion over the vaccine mandate status had resulted in missed opportunities to offer the vaccine to eligible youth and undermined public health messaging about the risks of HPV and the importance of vaccination.
Politics dictate public health	Community leaders reported that local elected officials’ political beliefs tend to shape public health initiatives and decision making. School officials who disagree with the mandate reportedly rebuff public HPV education and vaccination efforts offered by local health departments.
HPV as a public health blind spot	Public health officials and school employees reported an overall lack of HPV-related information disseminated by or through the school system. Local schools that are academically underperforming are primarily focused on maintaining their accreditation status, not engaging in “controversial” public health initiatives.

## Data Availability

According to IRB protocol data will not be shared with anyone outside the research team.

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
