# Peer review of "Perceptions of a State-Level HPV Vaccine Mandate and Exemption Option in Rural Virginia: A Qualitative Study"

_vaccines, 2024, doi:10.3390/vaccines12040401_

Round 1
Reviewer 1 Report
Comments and Suggestions for Authors
Introduction: Presents well and concise the literature gap. It would also benefit the literature gap to be presented through quantitative studies.
Lines 53-66 detailed well the existence of the only qualitative study; however, introducing the reader to some quantitative studies would be beneficial. The quantitative studies will also demonstrate the need to conduct this study.
Methods: There are major flaws in methods.
The authors state that the present manuscript is part of a larger mixed methods study. Thus, citing previous publications to avoid self-plagiarism is highly recommended.
The major flaw is the lack of a framework and how the study design was done. For example, the lack of references for the methodology is appalling. There are paragraphs without one reference.
The authors present “groups”; however, they state how the interview guide was designed. Since the authors use interchangeably these study designs, it will be beneficial to have a clear idea which methodology was used. To improve the flow of the manuscript, there is no need to have headings; however, here are some suggestions for a better flow.
a) Study design, including framework used for the interview guide.
b) Recruitment
c) Data collection, analysis, and rigor
Furthermore, the authors must cite the usage of snowball sample technique to avoid plagiarism.
Please remove the interview average length (line 80), which belongs to the results.
There are additional concerns in the methods: Which framework did the study use for data analysis? Please cite it and explain using rationale. Regarding the transcription, please clarify who transcribed the interviews and how bias was avoided. Furthermore, the was the coding done using software? If so, please reference it.
Results: There are major flaws in this section. A significant flaw is the number of participants in the study (n=40). The authors did not state why they selected this number (n=40) and how saturation was achieved.
There are major flaws in the presentation of the results. For example, the authors present the domains and then the themes, which are difficult to follow. It is highly recommended that the authors read other manuscripts and learn more on how to present qualitative data and how to accurately write about it so as not to skew the reader. The many different sub-themes and their own sub-themes were a bit confusing and hard to follow for the reader to remember which sub-theme went with which major theme.
Discussion: There are major concerns in this section. The discussion should start with a summary of the results. Lines 371-380 present the limitations of the study. It would be beneficial if the authors could expand on this section and combine it with the above paragraph that describes the strengths.
Author Response
Reviewer 1
“Lines 53-66 detailed well the existence of the only qualitative study; however, introducing the reader to some quantitative studies would be beneficial. The quantitative studies will also demonstrate the need to conduct this study.”
A brief discussion of additional papers that analyzed quantitative data pertaining to the Virginia HPV vaccine mandate has been incorporated into the penultimate paragraph of the introduction along with three additional citations.
“The authors state that the present manuscript is part of a larger mixed methods study. Thus, citing previous publications to avoid self-plagiarism is highly recommended.”
There are no other previous publications to come out of this study to cite at this time. We will be sure to cite this paper on any subsequent publications.
“The lack of references for the methodology is appalling. There are paragraphs without one reference.”
The original manuscript methods section contained 3 references, one in the first paragraph and two in the second citing our approach to sampling, thematic analysis, and inductive coding. We have expanded on our approach to thematic analysis and added an additional citation. It is unclear where additional references are being requested.
“The authors present “groups”; however, they state how the interview guide was designed. Since the authors use interchangeably these study designs, it will be beneficial to have a clear idea which methodology was used.”
The semi-structured interview guide was developed through a review of the relevant literature and discussion with subject matter experts. All participants were asked about knowledge, beliefs and attitudes concerning HPV and the HPV vaccine, as well as their perceptions of barriers and facilitators to vaccine uptake in their region. Clinicians were asked additional questions focused on their experience engaging with patients about HPV and the HPV vaccine in their clinical practice. Additional details explaining the development of the interview guide have been added to the methods section.
“To improve the flow of the manuscript, there is no need to have headings; however, here are some suggestions for a better flow: a) Study design, including framework used for the interview guide; b) Recruitment; c) Data collection, analysis, and rigor.”
We agree that the numbering system is a bit confusing, however we used the series of headings suggested by Vaccines in their manuscript template, which is provided on the journal website. If the editors decide there is flexibility in what headings are to be used, and the order in which they appear, we are happy to revise.
“The authors must cite the usage of snowball sample technique to avoid plagiarism.”
Snowball sampling is a common approach to qualitative sampling used in a wide variety of study applications. It does not require the use of a proprietary tool or instrument and does not reflect a unique idea or approach exclusively developed by any single research entity such that naming it in a paper without a citation could conceivably constitute plagiarism.
“Please remove the interview average length (line 80), which belongs to the results.”
The sentence describing average interview length has been moved to the results section.
“Which framework did the study use for data analysis? Please cite it and explain using rationale.”
Interviews were thematically coded using a mix of a priori concepts derived from the interview guide and an inductive immersion-crystallization approach to identify emergent themes. Additional details explaining this process and relevant citations were added to the methods.
“Please clarify who transcribed the interviews.”
Interview audio files were transcribed using Otter.ai, an AI-powered transcription software and then spot-checked for accuracy. We have clarified this in the methods section.
“Was the coding done using software? If so, please reference it.”
Our thematic coding did not use any specialized qualitative analysis software. We used Microsoft Word to organize, code, and analyze excerpts from our transcript data.
“A significant flaw is the number of participants in the study (n=40). The authors did not state why they selected this number (n=40) and how saturation was achieved.”
Given the exploratory nature of our study, we sought to include 9-17 subjects for each of our two primary subgroups, parent/guardian and clinician, and a convenience sample of other individuals (“community leaders”) with in-depth experience living and working in the community. Recent research suggests that this is sufficient to reach qualitative data saturation, although the precise amount varies widely depending on study goals, research questions, and the complexity of the sample population. During analysis we determined data saturation by assessing when the data in transcripts began to repeat itself, and when we felt we achieved a sufficient understanding of the identified themes (or codes) to address our primary research questions, thus rendering additional interviews redundant.
“There are major flaws in the presentation of the results. For example, the authors present the domains and then the themes, which are difficult to follow. It is highly recommended that the authors read other manuscripts and learn more on how to present qualitative data and how to accurately write about it so as not to skew the reader. The many different sub-themes and their own sub-themes were a bit confusing and hard to follow for the reader to remember which sub-theme went with which major theme.”
We added language clarifying that our results are organized into two primary domains, each containing several sub-themes. This will hopefully clarify our results for those unfamiliar with qualitative reporting stylistics.
We agree that the numbering system is a bit confusing, however we used the series of headings suggested by Vaccines in their manuscript template, which is provided on the journal website. If the editors decide there is flexibility in what headings are to be used, and the order in which they appear, we are happy to revise.
“The discussion should start with a summary of the results.”
We have added a brief summary of results in the first paragraph of the discussion.
“Lines 371-380 present the limitations of the study. It would be beneficial if the authors could expand on this section and combine it with the above paragraph that describes the strengths.”
We have expanded on the study limitations and combined this with the study strengths in the same paragraph.
Reviewer 2 Report
Comments and Suggestions for Authors
Comment: Is it possible to add the type of study (qualitative)? study) as part of the title to read as
Perceptions of a State-level HPV Vaccine Mandate and Exemption Option in Rural Virginia: A qualitative study
Methods
Comment: provide a justification of the sample size in each categories
The questions/questionnaire were prequalified
Author Response
Reviewer 2
“Is it possible to add the type of study (qualitative)? study) as part of the title to read as: Perceptions of a State-level HPV Vaccine Mandate and Exemption Option in Rural Virginia: A qualitative study”
That’s a great idea. We have revised the title accordingly.
“Methods: Provide a justification of the sample size in each categories”
Given the exploratory nature of our study, we sought to include 9-17 subjects for each of our two primary subgroups, parent/guardian and clinician, and a convenience sample of other individuals (“community leaders”) with in-depth experience living and working in the community. Recent research suggests that this is sufficient to reach qualitative data saturation, although the precise amount varies widely depending on study goals, research questions, and the complexity of the sample population (Hennink & Kaiser 2022). During analysis we determined data saturation by assessing when the data in transcripts began to repeat itself, and when we felt we achieved a sufficient understanding of the identified themes (or codes) to address our primary research questions, thus rendering additional interviews redundant.
Reviewer 3 Report
Comments and Suggestions for Authors
this is an important article. As HPV vaccine has the potential of eradicating cervical cancer, and its implementation is problematic, any effort to increase vaccination rates makes sense.
Some methodological comments: The demographics of the subjects could shed more light, as so many of the other states do not make vaccination mandatory, like have the subjects lived in Virginia for many years? The 3 interviewer guides should be available to readers, possibly as supplementary information. How many interviewers conducted the study? How many of the interviews were face to face? How many in each group declined participation?
In the discussion I miss 2 major points: One of the disputed in all democracies is the amount of government involvement (Autonomy versus enforcement). The study would gain additional value if participants were given feedback, like informing them about the results and conclusions of the study.
Author Response
Reviewer 3
“The demographics of the subjects could shed more light, as so many of the other states do not make vaccination mandatory, like have the subjects lived in Virginia for many years?”
Residence in Virginia was one of the inclusion criteria for our study, but we did not collect data on subjects’ length of time living in the state.
“The 3 interviewer guides should be available to readers, possibly as supplementary information.”
We used one primary interview guide, with additional modified questions for healthcare providers asking about their clinical practice. A list of interview topics asked about in the guide has been added to the methods.
“How many interviewers conducted the study? How many of the interviews were face to face? How many in each group declined participation?”
We added language clarifying that all study authors were involved in conducting the interviews, and that they were primarily held over Zoom to accommodate people’s schedules and covid precautions. Interviews were only conducted by those who expressed interest on a separate survey (not reported on here).
“One of the disputed in all democracies is the amount of government involvement (Autonomy versus enforcement). The study would gain additional value if participants were given feedback, like informing them about the results and conclusions of the study.”
Yes, we agree. We have plans to share study findings with the communities in the near future.
Round 2
Reviewer 1 Report
Comments and Suggestions for Authors
Although the manuscript improved and addressed most of my concerns, the methodology lacks the appropriate rigor. For example, the authors should explain the rationale for conducting a qualitative study based on scientific needs. The authors state “we thought” in lines 79-84. Additionally, the qualitative study needs robust information for the methodology and references. Since this study is part of a mixed methods, please consult Creswell et. al. to address regarding the study design. [1, 2]
Here are some examples on how to write a qualitative study.[3, 4]
While some quotes in the results section use quotation marks, others do not. Please be consistent. Additionally, please state which interview the quote belongs to.
1. Fetters, M.D., L.A. Curry, and J.W. Creswell, Achieving integration in mixed methods designs—principles and practices. Health services research, 2013. 48(6pt2): p. 2134-2156.
2. Creswell, J.W., et al., Advanced mixed methods research designs. Handbook of mixed methods in social and behavioral research, 2003. 209: p. 240.
3. Spiers, J., et al., The experience of antiretroviral treatment for Black West African women who are HIV positive and living in London: an interpretative phenomenological analysis. AIDS and Behavior, 2016. 20(9): p. 2151-2163.
4. Cernasev, A., et al., Importance of pharmacist-patient relationship in people living with HIV and concomitant opioid use disorder. Exploratory Research in Clinical and Social Pharmacy, 2021. 3: p. 100052.
Comments on the Quality of English Language
It has improved.
Author Response
Although the manuscript improved and addressed most of my concerns, the methodology lacks the appropriate rigor. For example, the authors should explain the rationale for conducting a qualitative study based on scientific needs. The authors state “we thought” in lines 79-84.
We have included a justification for the current study in the manuscript on lines 69-74. Namely, previous studies on HPV vaccine mandates have either been policy focused or exclusively drawn on quantitative data, leaving a qualitatively informed understanding of these issues largely unaddressed in the broader literature, and acutely so within the context of Virginia. Further, unlike previous studies, ours uniquely focused on geographical areas of low uptake where future public health and policy efforts are needed. We have included additional language clarifying this gap in the literature and how our qualitative findings can inform future studies.
As a point of clarification, we did not write “we thought” in lines 79-84. The sentence in question is copied here. “Given the exploratory nature of our study, we sought to include 9-17 subjects for each of our two primary subgroups, parent/guardian and healthcare provider, and a convenience sample of so-called community leaders with in-depth experience living and working in the community.”
Additionally, the qualitative study needs robust information for the methodology and references. Since this study is part of a mixed methods, please consult Creswell et. al. to address regarding the study design.
The data presented in this paper is indeed part of a larger mixed-methods study. However, this paper focuses exclusively on a portion of the qualitative data gathered during this larger effort. Future papers may present a mixed-methods analysis integrating the qualitative and quantitative data.
Here are some examples on how to write a qualitative study.
Thank you. We will certainly take your suggestions under advisement.
While some quotes in the results section use quotation marks, others do not. Please be consistent. Additionally, please state which interview the quote belongs to.
Quotes longer than 40 words are indented and center justified, rather than included in text with quotes, as is the convention in some publications. If the Vaccines journal has a different formatting requirement, we are happy to oblige. Additionally, we made sure that each quote used identifies which population group it came from. We have avoided using more specific identifiers in order to avoid inadvertently compromising our interviewees’ identities in these relatively small, close-knit rural communities.